## Original Research Article

*Acer saccharum*; sap extraction; tree-ring width; wood anatomy; xylem.

**Corresponding author:**
Hannah Grace McNulty;
Email: hgmcnulty@etu.uqac.ca

**Associate Editor:**
Aurélien Tellier

# To flow or to grow? Impacts of tapping on sugar maple

Hannah Grace McNulty[1] , Roberto Silvestro[1], Minhui He[1], Fabio Gennaretti[2,3] and Sergio Rossi[1]

[1]Laboratoire sur les écosystemes terrestres boreaux, Département des Sciences Fondamentales, Université du Québec à Chicoutimi, Chicoutimi, QC, Canada; [2]Institut de Recherche sur les Forêts, Groupe de Recherche en Écologie de la MRC-Abitibi, Chaire de Recherche du Canada en dendroécologie et dendroclimatologie, Université du Québec en Abitibi-Témiscamingue, Amos, Canada; [3]Department of Agricultural, Food and Environmental Sciences, Marche Polytechnic University, Ancona, Italy

## Abstract

Maple sugaring is a rapidly growing industry in North America. Maples are tapped annually, thus undergoing repeated wounding and resource reduction for sap water collection. We aim to understand the effects of tapping and sap exudation on annual radial wood growth and xylem traits in sugar maple (*Acer saccharum* Marsh.), utilizing eight mature trees monitored during 2018-2021 in Simoncouche, Canada. Compared to the first year of tapping, trees exhibited a 49.7% drop in tree-ring width. Vessel density, potential hydraulic conductivity and hydraulic vessel diameter decreased, but not lumen area. We showed evidence of a trade-off among sap extraction, resource depletion and reduced tree growth. The repeated reduction of resources through tapping can have a detrimental effect on tree growth, even if the effect on the hydraulic function remains marginal. These insights underscore the need for sustainable tapping practices that consider the long-term health and productivity of sugar maple trees.

## 1. Introduction

In Canada, maple syrup production holds a symbolic place and plays a significant economic and cultural role. Indigenous peoples introduced maple sap to the European explorers who were colonizing the Americas (Koelling *et al.*, 1996). This traditional knowledge has been passed down through the generations and has become an integral part of Canadian heritage. Since its introduction, the maple syrup industry has persevered through time and recently, experienced remarkable growth, supporting local economies through job creation and contributing to the economic stability and development of several regions (Duchesne et al., 2009; Whitney & Upmeyer, 2004). Today, the maple industry is an international market, with maple syrup consumed in more than 50 countries worldwide (Pires et al., 2021) with exports of maple products earning 616 million dollars for Canada (Government of Canada, 2023). Quebec is the world's top-producing area, accounting for about 77% of all maple syrup production in North America (Government of Canada, 2023). A near-constant increasing number of tapped trees is being driven by a combination of increasing demand for exportation and technological advancements in tapping (Rademacher et al., 2023). Although these innovations have significantly boosted production and efficiency, the impact of intensive tapping on the overall health and longevity of sugar maple trees remains uncertain, potentially jeopardizing the industry.

Traditional sap collection involved attaching a bucket to the tree trunk to gather sap exuded from freshly drilled holes (Lagacé et al., 2019; Rademacher et al., 2023). Sap exudation mainly occurs due to the positive stem pressures generated by freeze-thaw cycles (Cirelli et al., 2008; Copenheaver et al., 2014). In the 1960s, plastic tubing systems were implemented, and vacuum pumps started to be used to intensify sap flow and yield (Lagacé et al., 2019; Ouimet et al., 2021). Nowadays, vacuum levels of up to 95 kPa make the amount of sap collected per tree on average 2–3 times greater than the historical gravity-based methods (Isselhardt et al., 2016; Ouimet et al., 2021). When tapping a tree for sap extraction, the tap hole constitutes an injury that allows air and micro-organisms to enter, potentially creating a pathway into the sapwood

where bacteria could be fostered and harm the sapwood (Rademacher et al., 2023). To limit the risk of propagating foreign agents, the tree compartmentalizes the wound (Gibbs & Smith, 1973; Guillemette et al., 2023; Morris et al., 2019; Perkins et al., 2015; van den Berg et al., 2016). The wound eventually becomes a non-conductive part of the xylem, affecting water transport and future sap production (Perkins et al., 2015; van den Berg et al., 2016). For this reason, each season, the tree is tapped in a different spot, submitting the tree to repeated wounding (Chantuma et al., 2009). These repeated wounds are considered defects in lumber production, which diminish the manufacturing value of maple boards (Guillemette et al., 2023). Furthermore, the period of sap collection coincides with the time when maples rely on the natural mobilization of stored sugars in early spring (Dietze et al., 2014; Lagacé et al., 2019; Muhr et al., 2016; Ouimet et al., 2021), thus reducing the resources for the trees.

To date, questions remain as to whether, and how much, tapping and the consequent resource depletion from sap collection affect the growth process in maple. Investigations on tree ring width addressing the issue have shown contrasting results. Some studies reported a negative relationship between resource extraction through tapping and growth (Copenheaver et al., 2014; Isselhardt et al., 2016), while others remained inconclusive (Ouimet et al., 2021). These diverging results may depend on several factors such as the frequency and intensity of tapping, tree age and environmental conditions (Boakye et al., 2023; Ouimet et al., 2021), suggesting that maples in different geographical regions and environmental stressors may respond differently to tapping.

The response of wood anatomical traits in tapped trees is a relevant issue, as it may provide important information on changes in resource allocation and the hydraulic functioning of tapped trees. The current understanding of wood anatomical traits is largely dominated by studies in conifers (Chen et al., 2022), while broadleaves, and especially maple, remain partially neglected. For example, the anatomy of red maple in the northeastern United States demonstrates a high plasticity in relation to the length of the growing season, but seems to be unrelated to climate (Chen et al., 2022). Similarly, the hydraulic architecture of sycamore maple in the Mediterranean basin did not show any adjustment for efficiency or safety under climate variation, although a slight positive relationship between average vessel size and vessel density was found with precipitation (Rita, 2015). Although a solid relationship between the wood traits of maple and climate stressors still remains to be demonstrated, sap exudation may result in a different outcome because it is a constant stressor that could possibly induce resource reduction to a point in which growth is negatively impacted. Therefore, a detailed assessment of tapping effects on growth performance and wood characteristics is needed to guide sustainable practices and ensure the long-term viability of the maple syrup industry.

This study takes place at the northern limit of Maple, at the border of its ecological niche, where the individuals are most sensitive to disturbances. Such sites could provide valuable insights into carbon allocation dynamics and plant growth. We aim to understand the impact of tapping and sap collection through a gravity system on radial stem growth. We evaluate whether tapping alters xylem anatomical and functional traits, including vessel size and density, average lumen area and hydraulic conductivity, as these traits, to our knowledge remain partially unknown in the scientific literature. Accordingly, we test the hypotheses that (1) maple tapping leads to a reduced wood growth performance; and (2) tapping results in changes in xylem anatomical and functional

traits, specifically leading to a lowered efficiency of water movement within the xylem to increase structural support and water transportation safety.

## 2. Materials and methods

### 2.1. Study area

The study was conducted in a sugar maple (*Acer saccharum* Marsh.) stand at the Research forest of Simoncouche (48°15'N, 71°15'W), Quebec, Canada. The stand is located in a mixed forest including yellow and white birch (*Betula alleghaniensis* Britt. and *Betula papyrifera* Marsh.) and coniferous species, such as balsam fir (*Abies balsamea* (L.) Mill.) and white spruce (*Picea glauca* (Moench) Voss.). Climatic conditions are typically boreal, characterized by cold winters with absolute minimum temperatures of −34 °C and short cool summers, with absolute maximum temperatures occasionally reaching 33 °C. The mean annual temperature is 2.2 °C. Total precipitation is 1382 mm, of which 131 mm falls as snow from November to April (Rossi et al., 2011). The site is situated on well-drained podzolic soil with an organic layer of 10 cm in thickness.

### 2.2. Experimental design and tapping

The study was conducted on eight co-dominant, healthy, mature sugar maples that had not previously been tapped. In spring 2018, four trees were selected for tapping and four were used as an untapped control. The trees were tapped annually for the duration of the sugar seasons until 2021, for a total of four years. Tapped trees were >20 cm in diameter at breast height (avg. 26 cm, DBH, Supplementary Table S1), and only one taphole was used per year following best practices for maple syrup production (van den Berg et al., 2016) and Quebec government regulations (Gouvernement du Québec, 2018). Control and tapped trees have an average diameter of 23.6 cm and 28.4 cm, respectively (DBH, Supplementary Table S1). The tap holes, 0.8 cm (5/16 inch) in diameter and 3.5–3.8 cm in depth were equipped with a traditional tubing spout used for the gravity sap collection system (Kurokawa et al., 2024).

### 2.3. Wood cores and data collection

In October 2023, after radial growth cessation, we collected four wood cores at breast height from each tree using a 5 mm increment borer (Haglöf, Sweden). The cores were collected vertically at least 12 cm apart from the tap holes to avoid interference with non-conducive wood columns associated with prior tapholes. The cores were airdried, cut to size, glued into wooden blocks and prepared for ring-width analysis by sanding, polishing and applying chalk to enhance visibility. All samples contained the previous 8–10 tree rings, which were used to date the growth years accurately. This study only focused on years the 2018–2021. The tree-ring widths were measured using a LINTAB 6 (Rinntech, Victoria, Canada).

### 2.4. Anatomical measurements

After the tree-ring width measurements, the cores were boiled in water and sectioned using a rotary microtome into transverse sections 13–15 μm in thickness, stained with safranin for 5 minutes, rinsed in ethanol, and mounted using Eukitt Quick-hardening

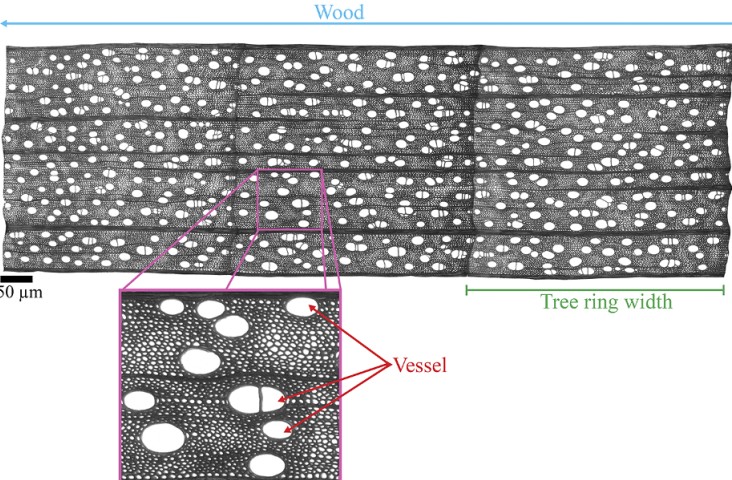

**Figure 1.** Cross sectional view of a tree core of a studied sugar maple tree compiled from images taken at ×5 magnification. The inset shows an actual tree ring segment where vessel measurements were taken. Ring length is uniform in the radial direction for all tree cores. Scale bar = 50 μm.

mounting medium. We collected anatomical images of each tree ring at ×5 magnification and measured the anatomical traits using WinCELL V.2019e (Regent Instruments Inc., Quebec, Canada) (Figure 1). Measurements included vessel density, number of vessels, cross-sectional vessel lumen area and area of the tree ring per each complete tree ring segment with a uniform tangential dimension. We also measured the relative horizontal and vertical positions of each vessel within the tree ring to describe the variation in vessel lumen area trend across the tree ring. Vessel density utilized a standardized area excluding the late-wood formation portion of the ring. A loess function was applied to obtain continuous values across the tracheidogram.

Vessel diameter was calculated from the lumen area, assuming a perfect circular shape. Two different measurements of hydraulic performance were computed, the hydraulic vessel diameter ($D_h$, μm), i.e., the average diameter of vessels contributing most to water movement, and the potential hydraulic conductivity ($K_p$, Kg m$^{-1}$ MPa$^{-1}$ s$^{-1}$), i.e., the ability with which a plant can move water through the stem. The hydraulic vessel diameter within the ring was calculated by the following formula (Buttó et al., 2021; Sperry et al., 1994):

$$D_h = \frac{\sum_{i=1}^{n} d^5}{\sum_{i=1}^{n} d^4} \qquad (1)$$

where $d$ is the vessel diameter and $n$ is the number of vessels in the tree ring portion. The theoretical conductivity, $K_h$ (m$^3$ s$^{-1}$), of the tree ring portion was calculated by first adding up the conductance of all vessels from Hagen–Poiseuille's equation:

$$K_h = \frac{\pi \times \rho \times \Sigma D^4}{128 \times \eta} \qquad (2)$$

where $\rho$ is the density of water (998.2 kg m$^{-3}$), and $\eta$ is the viscosity of water (1.002 10$^{-9}$ MPa s), both at 20 °C (Cruiziat et al., 2002). Then, potential conductivity, $K_p$, was calculated by the following formula (Zimmermann et al., 2021):

$$K_p = \frac{K_h}{A} \qquad (3)$$

where $K_h$ is the calculated theoretical hydraulic conductivity, and $A$ is the area of the analyzed tree ring from the corresponding wood segment.

### 2.5. Statistical analyses

The effects of tapping on tree ring width and anatomical traits were assessed by mixed models, with the year and tapped status being included as fixed effects. The cores nested within the trees were considered as a random effect. We analyzed the effect of tapping, year, and their interaction on tree-ring width and anatomical traits using a Tukey's HSD test for multiple comparisons. We also used a principal component analysis (PCA) to verify the association between growth and anatomical and functional traits. All statistics were performed in R version 4.1.2 (R Core Team, 2021) and JMP Pro 17 (JMP Statistical Discovery LLC, 2022).

### 3. Results

### 3.1. Tree-ring width

In 2018, the initial year of experimentation and the first year of tapping, our analysis revealed no significant difference in tree-ring widths between treatments (tapped and untapped trees) (Figure 2, Supplementary Figure S1). In 2019, tree cores from untapped trees

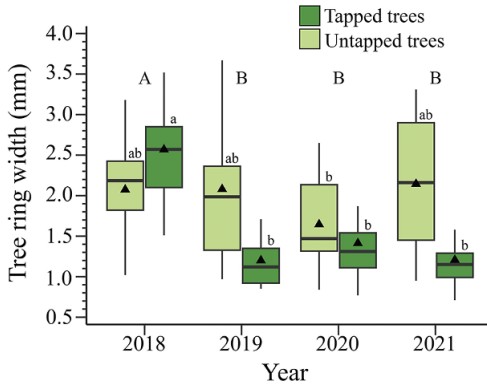

**Figure 2.** Tree ring width in tapped and untapped sugar maples during the four study years in Simoncouche, Quebec, Canada. The boxplots represent upper and lower quartiles, with the whiskers indicating the 10th and 90th percentiles. The horizontal black line represents the median and the black triangles represent the average. According to posthoc analysis, tapping treatment was significant, the lowercase letters indicate differences in the interaction between year and treatment, and the capital letters indicated differences among years.

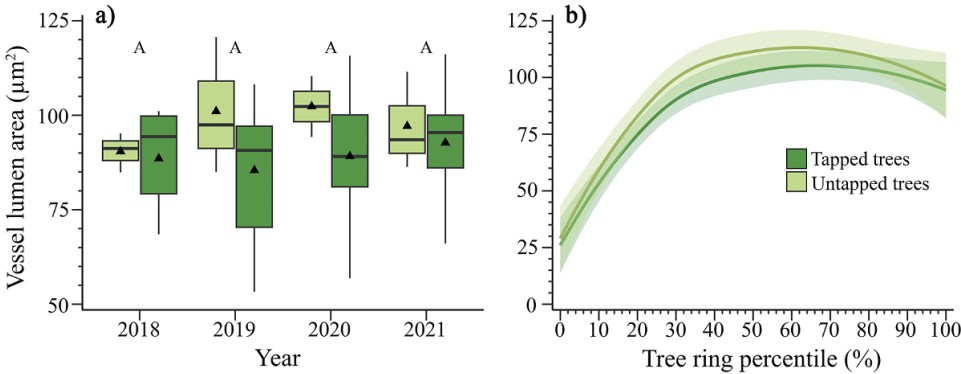

**Figure 3.** (a) Vessel lumen area in tapped and untapped maples in Simoncouche, Quebec, Canada. The boxplots represent upper and lower quartiles, with the whiskers indicating the 10th and 90th percentiles. The horizontal black line represents the median, and the black triangles represent the average. The capital letters indicate differences among years, tapping treatment and year were not significant according to posthoc analysis. (b) Trend of average vessel lumen area across tree rings in tapped and untapped maples in Simoncouche, Quebec, Canada. The trend lines result from a loss function (span 1.2), with the respective color backgrounds representing 95% confidence intervals of the loess.

had a total average tree-ring width of 2.01 mm, while the tree-ring width in tapped trees reached 1.21 mm, showing a 54.4% decrease compared to the previous year (Figure 2, Supplementary Figure S1). In the next two years (i.e., 2020 and 2021), tree-ring width in both untapped and tapped trees remained consistent with the measurements in 2019 (Figure 2, Supplementary Table S2, Figure S1). Overall, after the first year of tapping in 2018, tapped trees exhibited a 49.7% drop in tree-ring width (Supplementary Table S2, Figure 2, Supplementary Figure S1).

### 3.2. Vessel lumen area

On average, no differences in vessel lumen area were observed between years, treatment, or their interaction ($p > 0.05$, Supplementary Table S3, Figure 3a). We observed a larger variation in the size of vessels in tapped trees. Specifically, tapped trees showed an average lumen area ranging from 53.2 to 148.7 $\mu m^2$, while the lumen area of untapped trees ranged from 84.9 to 128.8 $\mu m^2$ (Figure 3a). The vessel size throughout the tree rings showed similar trends between tapped and untapped trees, although the average lumen area was slightly higher in untapped trees (Figure 3b).

### 3.3. Hydraulic conductivity traits

Tapping affected vessel density in the stem across years, and post-hoc tests showed significant variations in the interaction of year and treatment ($p < 0.05$, Supplementary Table S4, Figure 4). In untapped trees, vessel density showed steady incremental increases over the years, starting from a density of 525 vessels mm$^{-2}$ in 2018 to 767 vessels mm$^{-2}$ in 2021 (Figure 4). In tapped trees, the density remained more constant, ranging from 431 vessels mm$^{-2}$ in 2018 to 514 vessels mm$^{-2}$ in 2021 (Figure 4).

Hydraulic vessel diameter showed significant variation for the interaction tapping × year. ($p < 0.05$, Supplementary Table S6). In 2018, the first year of tapping, no significant variations between tapped and untapped trees were observed, with hydraulic vessel diameters of 12.6 $\mu m$ and 13.3 $\mu m$, respectively (Figure 4). In 2019, the average hydraulic vessel diameter of tapped trees was statistically lower than that of untapped trees, including the significant interaction year × tapped status ($p < 0.05$, Supplementary Table S6, Figure 4). The results of our model exhibited a significant effect of tapping on potential hydraulic conductivity ($p < 0.05$, Supplementary Table S5, Figure 4). The potential hydraulic conductivity was

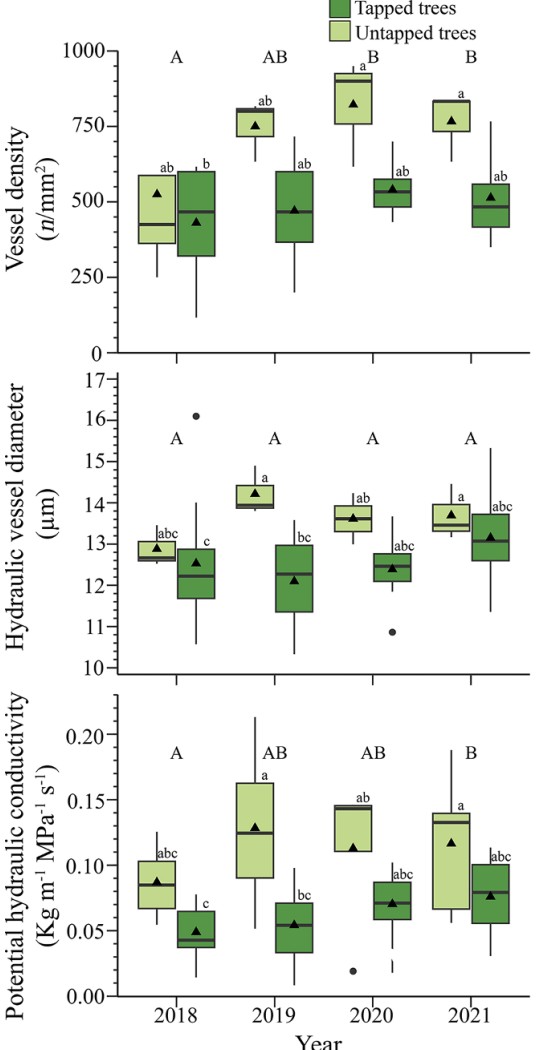

**Figure 4.** Hydraulic conductivity within the stem in tapped and untapped maples in Simoncouche, Quebec, Canada. The boxplots represent upper and lower quartiles, with the whiskers indicating the 10th and 90th percentiles. The horizontal black line represents the median, and the black triangles represent the average. According to posthoc analysis, tapping treatment was significant for potential hydraulic conductivity, the lowercase letters indicate differences in the interaction between year and treatment, and the capital letters indicate differences among years.

0.06 and 0.11 kg m$^{-1}$ MPa$^{-1}$ s$^{-1}$ in tapped and untapped trees in 2018, respectively. We observed a gradual increase of 0.01 kg m$^{-1}$ MPa$^{-1}$ s$^{-1}$ per year in the potential hydraulic conductivity in tapped trees over the study period, starting from 0.06 kg m$^{-1}$ MPa$^{-1}$ s$^{-1}$ in 2018 to 0.09 kg m$^{-1}$ MPa$^{-1}$ s$^{-1}$ in 2021 (Figure 4). Notably, there was a large amount of variation after the first year of tapping with the widest range detected in 2019 from 0.008 to 0.213 kg m$^{-1}$ MPa$^{-1}$ s$^{-1}$ (Figure 4).

### 3.4. Principal component analysis

The relationships between anatomical and hydraulic traits were analyzed in detail by performing a PCA (Supplementary Figure S2). The two components explained 80.8% of the variance in the studied variables. The first principal component (PC1) explained 61.6% of the overall variance and was positively associated with vessel density, potential hydraulic conductivity, hydraulic vessel diameter, vessel lumen area and had a marginal influence on tree-ring width. The second principal component (PC2), explaining 19.2% of the overall variance, was positively correlated with tree-ring width, potential hydraulic conductivity and vessel density (Supplementary Figure S2). PC2 was also negatively correlated with hydraulic vessel diameter and vessel lumen area. The distribution of the groups is heterogeneous, with tapped trees being more clustered in the bottom left quadrant, while the untapped trees are more spread out due to a higher variability.

## 4. Discussion

This study investigated the impact of gravity-based tree tapping on radial growth and anatomical and functional traits of xylem in sugar maple growing at the northern edge of its natural range in Quebec, Canada. Compared to the control, the trees tapped annually for sap collection exhibited a decrease in tree-ring width after the first year of sap production. The limited number of sugar maple trees available for study in this area may result in an underestimate of the overall impact. Tree tapping had compounding effects on the overall conductivity, resulting from a lower vessel density, hydraulic vessel diameter and potential hydraulic conductivity. These results support the initial hypothesis that tapping leads to reduced growth performance and stimulates anatomical responses in wood features within the stem that affect the functional performance of the xylem.

### 4.1. Tapping and tree ring width

Tapped trees showed reduced annual radial growth after successive tapping years compared to untapped trees. Tapping is a human-induced disturbance to maples the effects of which are not fully understood, particularly regarding long-term growth patterns (Ouimet et al., 2021). During the first growing season after tapping, in 2018, tapped and untapped trees showed comparable growth. However, a reduction in growth was observed in all tapped trees from the second year of the treatment. This finding aligns with aspects of previous studies showing declines in growth performances after the first tapping year, or even later (Copenheaver et al., 2014; Isselhardt et al., 2016; Ouimet et al., 2021).

The significant reduction in tree-ring width observed after the first year of tapping may indicate that tapping induces considerable stress on maple. During the active photosynthetic period, from leaf unfolding in spring to senescence in fall, atmospheric carbon is converted into carbohydrates (Muhr et al., 2016). These sugars support important sinks such as primary and secondary growth and sustain a number of metabolic processes, and finally, a fraction is stored during dormancy. Non-structural carbohydrates are kept in the parenchymatous tissues of wood and bark throughout winter (Chantuma et al., 2009; Dietze et al., 2014; Von Arx et al., 2015). During dormancy, trees rely on starch reserves for survival and cold tolerance (Wong et al., 2005). In late winter, these stored carbohydrates are mobilized in the form of soluble sugar, increasing sap sugar concentrations to 1–6% (Dietze et al., 2014; Lagacé et al., 2019; Muhr et al., 2016; Ouimet et al., 2021). In this framework, tapping practices affect the tree storage pool which sustains the sinks of carbon until photosynthesis from the new leaves is able to satisfy the carbon needs of the tree (Dietze et al., 2014; Hartmann & Trumbore, 2016; Muhr et al., 2016). In detail, the decline in growth performance in tapped trees is likely due to the effect of tapping on the balance between carbon storage and immediate usage (Copenheaver et al., 2014; Isselhardt et al., 2016), and this impact emerges more clearly in the second year of tapping when the carbon reserves have been significantly depleted.

Our results align with other studies from other productions showing that the collection of resins from Masson pine (Chen et al., 2015), Aleppo pine (Papadopoulos, 2013) and Maritime pine (Génova et al., 2014), as well as the tapping of rubber trees for latex production (Chantuma et al., 2009; Silpi et al., 2006), typically results in a sharp decrease in the trees' radial growth. At the beginning of spring, sugar is mobilized and translocated, providing the necessary energy and resources for bud break and the reactivation of growth (Lagacé et al., 2019; Ouimet et al., 2021). In this moment sap collection also takes place, likely intercepting soluble sugars during their translocation, forcing the tree to rely on older stored carbohydrates. This affects the overall carbon storage, which is likely subsequently replenished with freshly assimilated carbon at the expense of wood production.

Sap collection could potentially also affect the overall fitness of tapped trees, leading to a negative impact on tree crown vigor (Copenheaver et al., 2014). A reduction in the crown size contributes to a lower photosynthetic rate and, consequently, reduced carbon assimilation (Copenheaver et al., 2014). Additionally, the wound created by the tapping requires a mobilization of resources to be repaired which could be a significant factor affecting the growth performance (Copenheaver et al., 2014). It is still challenging to identify a precise cause for the observed reduction, and it is likely that all of these factors contribute, to some extent, to the overall decline in growth after the first year of tapping. This uncertainty highlights the need for a much more in-depth exploration of the effects of tapping on tree fitness and performance, especially to gain a clearer understanding of the long-term impacts of tapping on maple trees.

### 4.2. Tapping and hydraulic conductivity

Tapped trees exhibited a decrease in potential hydraulic conductivity due to reduced vessel density, without significant differences in the vessel's morphology characteristics. In particular, vessel lumen area showed no change in size based on treatments or years. It is well known that vascular plants can adjust the anatomy of vessel conduits under differing conditions, making vessels larger or smaller, to protect against embolism and meet physical strength needs (Carrer et al., 2015; Lens et al., 2011; Zanne et al., 2010). However, maple species seem unable to respond to stressors through modifications in vessel size (Lens et al., 2011). This observation also aligns with what is recorded in Mediterranean maples, which

showed no significant variation in vessel size when exposed to drought (Rita, 2015).

Despite the lack of significant variations in vessel area, our results showed that the hydraulic vessel diameter was lower in tapped trees compared to the control. The hydraulic vessel diameter measures the average diameter of vessels contributing most to water movement (Cruiziat et al., 2002; Sperry et al., 1994). For this reason, this trait is sensitive to small changes in vessel size of larger vessels (Sperry et al., 1994). Although closely linked, it is important to note that hydraulic vessel diameter only considers the vessels that are contributing to water movement, thus excluding the small vessels from the calculation (Cruiziat et al., 2002). The hydraulic vessel diameter showed a significant difference between tapped and untapped trees. The years 2019 and 2020 show a difference in vessel size with tapped trees having the smaller hydraulic vessel diameter. Moreover, hydraulic vessel diameter is driven by an allometric relationship with tree size (Buttó et al., 2021; Carrer et al., 2015), and therefore remains relatively constant considering the low variability in untapped trees. Conversely, we find that tapping affects the size of the tree, and the hydraulic diameter remains more stable when compared to untapped trees. The stable hydraulic vessel diameter represents a shift to decreased mechanical strength and increased efficiency within the stem during those years (Lens et al., 2011; Zanne et al., 2010).

Tapping induces a reduction in vessel density, which contributes to the observed decrease in potential hydraulic conductivity in tapped trees over the study years. In our study, vessel density remained consistent, not increasing the density of vessels per section area across years in tapped trees. In contrast, vessel density increased per year in untapped trees when new, thicker rings, had more vessels contributing to hydraulic conduction. The link between increased vessel density and increased xylem conductivity is well demonstrated (Rita, 2015). Variations in wood anatomical characteristics, or in this case, the lack of the expected variations may represent an adaptive solution to balance hydraulic efficiency with structural support (Fonti et al., 2010). Likely, tapped trees under stress prioritize mechanical stability, while sacrificing hydraulic efficiency.

### 4.3. To flow or to grow?

Our results show that trees tapped for sap collection exhibited a reduction in growth performance compared to untapped trees. This study was performed by observing the effect of tapping utilizing a gravity tapping system. However, to maximize yield, modern producers are equipped with vacuum systems (Isselhardt et al., 2016; Lagacé et al., 2019; Ouimet et al., 2021), that can yield twice as much sap as gravity extraction (Lagacé et al., 2019). Considering the significant reduction in growth with gravity sap collection shown in this study, utilizing vacuum tapping and doubling the extraction rate could potentially further impact the growth performance of tapped trees. There is a need for further studies to assess tree growth performance and, more broadly, the overall carbon budget of trees subjected to sap collection, with or without vacuum systems, to better understand the impact of this practice on maple fitness. Specifically, to account for variability in growth performance among trees (Silvestro et al., 2022), future research should employ larger sample sizes, and, considering interannual variability, analyze tree performance on longer time scales. While it can be challenging to identify a large group of trees tapped simultaneously and in which the first tapping date span multiple decades, such experimental design could provide more definitive insights into the impact of tapping on growth performance. Nevertheless, the vacuum technique has some physiological advantages, such as smaller diameter spouts reducing the amount of internal compartmentalization (Perkins et al., 2015), and more rapid repair of the taphole (Staats & Kelley, 1996). However, further studies investigating the growth performances and more in general the global carbon budget of trees submitted to sap collection with or without vacuum systems are needed in order to understand the impact of this practice on maple fitness.

This study takes place at the northern limit of the maple range in northeastern North America. Previous studies assessing the impact of tapping on maple tree growth across latitudinal gradients have yielded contrasting results, demonstrating that the relationship between tapping practices and reduced growth still needs deeper investigation (Copenheaver et al., 2014; Ouimet et al., 2021). It is likely that at higher latitudes of the maple syrup production area, the shorter growing season, decreased photosynthetic activity, and cooler temperatures contribute to the reduced growth performance compared to maple trees in warmer climates (Hartmann & Trumbore, 2016). Future research should investigate multiple sites, ideally under divergent climatic conditions, to also assess how the local environment influences the relationship between resource extraction through tapping and tree growth processes. This study highlights the urgent need for further research into the long-term impacts of tapping with a vacuum system, particularly under varying climatic conditions. These pieces of evidence are important factors in potentially limiting the exploitation of sap production in the northern regions. While innovations like vacuum systems have significantly boosted production, their effects on the overall health and longevity of maple remain uncertain. Therefore, gaining new insights into the impact of tapping on growth performance is essential to refine current extraction practices while ensuring the long-term sustainability of the industry.

**Open peer review.** To view the open peer review materials for this article, please visit http://doi.org/10.1017/qpb.2025.9.

**Supplementary material.** The supplementary material for this article can be found at http://doi.org/10.1017/qpb.2025.9.

**Data availability statement.** Data associated with this paper will be available in Borealis, the Canadian Dataverse Repository, once accepted for publication.

### Acknowledgements

The authors thank P. Nadeau, S. Kurokawa, and F. Gionest for technical support, and A. Garside for checking the English text.

**Author contributions.** H.M., S.R. and R.S. conceived and designed the study; H.M. and S.R. conducted data gathering; H.M. analyzed the data and led the writing of the manuscript; S.R., R.S., F.G., M.H., contributed critically to the drafts; All authors have final approval for publication.

**Funding statement.** This work was funded by the Ministère des ressources naturelles et des forêts du Québec, Fonds de Recherche du Québec - Nature et Technologies (AccFor, project #309064), the Natural Sciences and Engineering Research Council of Canada (research programs Alliance and Discovery grants), Forêt d'Enseignement et de Recherche Simoncouche, the Syndicat des producteurs des bois du Saguenay-Lac-Saint-Jean, the Producteurs et productrices acéricoles du Québec, the Centre ACER.

**Competing interest.** The authors declare no competing interests.

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
