## [Reviewer Report]

This study deals with a subject of considerable interest to forest researchers and maple producers in both Canada and the U.S. It suffers from several important issues, particularly in the area of methodology. Some of this might be able to be addressed by more detailed descriptions in the methods section (especially in the characterization of trees and how tapping and tree core collections were done), but others may require deeper analysis if the information is available. Why were sap volume and sap sugar content not reported if collected? Why were tapping depths so different and shallow? What directions/azimuths was tapping and coring done at and were these the same for all study trees. Furthermore, some of the literature is cited inappropriately and the authors should find more appropriate references or should alter their statements to better represent what the referenced literature actually says. Lastly, it would help greatly to include growth rates from before the study was initiated and continue to show the growth data after cessation of sap collection and to place the growth rates in this study in context of those found in other studies, particularly given the site location at the edge of the sugar maple/sap collection range.

I have many a number of comments directly in the manuscript.

---

## [Reviewer Report]

Dear authors,

your study is based on a solid dataset, well written and I have only some minor comments. Please find them below.

1) Title: To bleed or to grow?

2) Line 57: is this really the case? Not simply from root pressure?

3) Line 62: you may provide a photo?

4) Line 65: you may cite one of Shigos (CODIT) works and Morris et al. 2020 (https://doi.org/10.1093/aob/mcz138)

5) Lines 94-100: you may add why lower growth is not appreciated, e.g. lower carbon fixation, or lower wood quality (if used for timber), lower timber yield etc.

6) Line 109: precipitation is quite high!

7) Lines 114-120: please add some information about the trees, such as height, diameter and age

8) Figure 1: please delete „xylem“ and „bark“, both are not necessary, and I would use „wood“ rather than „xylem“, and there is no bark present

9) Line 145: the scale bar can hardly be seen

10) Line 154: please also add the unit

11) Line 160: please also add the unit

12) Line 164: I am sure that there are older works where this formula was already introduced

13) Line166: please also add the unit

14) Line 169: what is meant with the corresponding tree segment? The analyzed tree ring from the wood core?

15) Line193: capital letters indicate differences among years for pooled non-treated and treated trees?

16) Line 211: Theoretical hydraulic conductivity or Potential hydraulic conductivity

17) Lines 217-228, present the results in order of the method description, hydraulic diameter before theoretical conductivity

18) Lines 218-219: add that this was the case in 2018

19) Figure 4: move the middle plod down, so that hydraulic vessel diameter is in the middle

20) Figure 4: do you have an idea why vessel density and vessel diameter and theoretical conductivity was increasing after 2018 in the non-treated trees: age effect or was 2018 a very dry year?

21) Line 321: ….vessels that are contributing to the bulk of water movement….meaning the smaller vessels also contribute to water flow

22) Lines 322-323: no space between paragraphs – it hinders the logical flow

23) Line 324: any explanation why there were no differences in 2021?

24) Line 334: do you have an explanation for this? Is this an age effect or because 2017 and/or 2018 was a drier year? I think you should drop a message here!

25) Lines 329-330: I do not understand the logics behind that message – in general and when related to the previous sentence

26) Page 17, would show this Figure in the main document, but the black fonts are not easy to read with the dark green background

---

## [Editor Report]

Dear authors,

Both reviewers and myself like the study topic and the manuscript. However, reviewer 1 has a number of concerns regarding the methodology and data analysis, which prevent acceptance at present. I encourage you to thoroughly and substantially revise the manuscript to provide the requested information as well as additional data if possible.

Best regards,

---

## [Reviewer Report]

Although the topic is of considerable interest to scientists, forest managers, and maple producers and the authors have done a lot to correct many of the errors in the manuscript, my strong concerns regarding methodology remain. There is no quantification of carbohydrate extraction rates, the diameters (and likely the ages) of the trees in the two groups are dissimilar, and the ring widths prior to (or after) sap extraction were not included in the analysis. These drawbacks leave considerable uncertainty as to the results and it doesn’t seem like it would require a great deal more effort to include this information. Secondly, despite there being a lot of valuable information from the 1998 Ice Storm on maple tree growth and carbohydrate relations, this was not included in the discussion, whereas examples from resin and latex production, which occur in a different part of the world (Mediterranean vs Eastern Canada), different types of trees (pine vs maple), different parts of trees (bark vs wood), and different abilities to redistribute resources (resin not able to be redistributed vs sugars in maple sapwood can be) were. Taken together, I advise rejection of the manuscript. I’ve included comments directly on the PDF that the authors might find useful.

Although the topic is of considerable interest to scientists, forest managers, and maple producers and the authors have done a lot to correct many of the errors in the manuscript, my strong concerns regarding methodology remain. There is no quantification of carbohydrate extraction rates, the diameters (and likely the ages) of the trees in the two groups are dissimilar, and the ring widths prior to (or after) sap extraction were not included in the analysis. These drawbacks leave considerable uncertainty as to the results and it doesn’t seem like it would require a great deal more effort to include this information. Secondly, despite there being a lot of valuable information from the 1998 Ice Storm on maple tree growth and carbohydrate relations, this was not included in the discussion, whereas examples from resin and latex production, which occur in a different part of the world (Mediterranean vs Eastern Canada), different types of trees (pine vs maple), different parts of trees (bark vs wood), and different abilities to redistribute resources (resin not able to be redistributed vs sugars in maple sapwood can be) were. Taken together, I advise rejection of the manuscript. I’ve included comments directly on the PDF that the authors might find useful.

Line 30. I don’t think you have adequately demonstrated that tapping “depleted” the resources of the tree. Although you measured sap volume yield, you did not present any of these data and did no measure sap sugar content. Alternatively, you could have measured carb content (sugars and starches) in the rings before and after tapping to demonstrate whether sap flow “depleted” resources.

Line 83. The relationship is not between tapping and growth, but between resource (carb) extraction and growth.

Line 122. Maple growth is strongly impacted by two key factors, competition for sunlight and water availability during the growing season. The supplemental data is a bit worrisome in that three out of four of the smallest trees were in the control group. The average dbh of the groups was 4.75 cm (1.9 inches) apart, which is quite sizable. Given the avg growth rates (and assuming avg growth over time), the control trees would be an avg of 40 yrs younger, and up to 75 yrs younger than the oldest tapped trees.

Secondly, although all the trees were in the same general area, it might be interesting to see the drought history at the study site over the study period. Larger trees (or tapped trees) might be more susceptible to low moisture availability than control trees. Your xylem characteristics measurements might also reflect this.

Line 133. Although I understand that another study looked at short term flow patterns, why is it not possible to tell us at least in summary how much sap was collected over each season so we have some perspective on the amount of carb resources extracted that were extracted from the tree? The lack of sap sugar content measurements is a serious oversight.

Line 197. I am still very uneasy with the thought that you did not measure ring width in the two groups of trees PRIOR to tapping. This might have quite easily shown that growth in the two groups was similar before tapping. You note in the methods that “All samples contained the previous 8-10 tree rings...”,

meaning you had rings going back to at least 2013-2015 (several yrs prior to tapping). So the data is there...why not use it? Similarly, you had rings from 2022 and 2023...what did these show? Presumably if carb resources were being depleted so quickly we might expect to see a similarly rapid recovery in ring width?

Line 204. Figure 2. Given the sizeable differences in dbh distribution between the tapped and untapped trees, basal area increment (BAI) would probably be a far better growth assessment indicator.

Line 284. Isselhardt observed reduced radial growth in the year of tapping. Your results showed that growth reductions were delayed a year. Any idea why?

Line 300. This discussion ignores the fact that carbs in maple stems are readily available in most cases for decades. As long as the xylem is functional (sapwood), sugars can be mobilized and moved to other sinks for a long time. Sugars produced and stored in the rings of one particular year do not necessarily stay there. Studies have shown that sugar from a taphole span a wide range of growing season, with an average sugar molecule in syrup being around 3 yrs old. This indicates that sugar is moved around in the stem and thus presumably available in years where there is a deficiency (drought) or demand (seed production). This also means that any single perturbation in production (or loss through sap flow from a taphole) will tend to be muted.

Line 304. Not sure how pertinent these references are. Resin and latex are derived from tree bark, not from xylem, and thus are not readily redistributable for other carbohydrate sinks like carbs in the sapwood (functional xylem) of maple stems are.

Line 379. It is also quite telling that maple the thousands of maple producers in the U.S. and Canada have NOT observed large growth reductions similar to those found in this study. It seems unreasonable that impacts of this magnitude would: 1) go unnoticed and 2) not have a cumulative impact over decades that would affect syrup production.

Line 379. It should have been quite easy to look through the copious literature available on: - the January 1998 ice storm to look for impacts on carb production and its implications on growth in sugar maple trees. These studies would be much more comparable than those looking at pine resin production in the Mediterranean - examine the typical growth rates in the area of study to see how representative the results from this study are.

---

## [Reviewer Report]

Dear authors,

the revision was done with care and almost all suggestions were addressed and if not, it was explained why. I still have, however, one final remark, where the changes (or no change) are unsatisfactory to me. The header “Hydraulic conductivity” in line 228 is misleading since it implies that hydraulic measurements (flow experiments) were performed, which was not the case. Anatomical measurements are only a raw proxy for the hydraulic conductivity and do not provide information on the actual conductivity, since resistances such as pits are not included. I would thus suggest as a header “Hydraulic conductivity traits” or “Hydraulic conductivity related anatomical traits”, if you do not want to write “Theoretical hydraulic conductivity” or “Potential hydraulic conductivity”.

---

## [Editor Report]

Dear authors,

As you can see the reviewers valued your efforts to improve the manuscript and agreed that the revised version is much improved. Both reviewers have some further points of improvement, which should be easily fixed (reviewer 1 also provided an annotated pdf). However, reviewer 1 has one critical point regarding the absence in your analyses of quantities of carbohydrate extraction rates, the diameters (and likely the ages) of the trees and the ring widths prior to (or after) sap extraction. This point needs to be addressed carefully if the manuscript is to be accepted. This point needs to be addressed in your revised version by either providing these additional analyses or discussing the limitations of your results resulting from the lack of such analyses. This is important to demonstrate the robustness of your study results and support the conclusions.

Best regards,

Aurelien Tellier

---

## [Editor Report]

Dear authors,

Thank you for providing the revised version and for addressing all points raised by the reviewers. I am happy to accept this new version of the manuscript for publication at QPB.

Thank you for choosing QPB.

Best regards

Aurelien Tellier